# Androgen Receptor in Breast Cancer—Clinical and Preclinical Research Insights

**DOI:** 10.3390/molecules25020358

**Published:** 2020-01-15

**Authors:** Aristomenis Anestis, Ilianna Zoi, Athanasios G. Papavassiliou, Michalis V. Karamouzis

**Affiliations:** 1Molecular Oncology Unit, Department of Biological Chemistry, Medical School, National and Kapodistrian University of Athens, 11527 Athens, Greece; arisanestis@gmail.com (A.A.); ilianna.zoi@gmail.com (I.Z.); papavas@med.uoa.gr (A.G.P.); 2First Department of Internal Medicine, Laikon General Hospital, Medical School, National and Kapodistrian University of Athens, 11527 Athens, Greece

**Keywords:** androgen receptor, breast cancer, triple negative breast cancer, estrogen receptor, androgens, antiandrogens

## Abstract

The Androgen Receptor (AR) is emerging as an important factor in the pathogenesis of breast cancer (BC), which is the most common malignancy among females worldwide. The concordance of more than 70% of AR expression in primary and metastatic breast tumors implies that AR may be a new marker and a potential therapeutic target among AR-positive breast cancer patients. Biological insight into AR-positive breast cancer reveals that AR may cross-talk with several vital signaling pathways, including key molecules and receptors. AR exhibits different behavior depending on the breast cancer subtype. Preliminary clinical research using AR-targeted drugs, which have already been FDA-approved for prostate cancer (PC), has given promising results for AR-positive breast cancer patients. However, since the prognostic and predictive value of AR positivity remains uncertain, it is difficult to identify and stratify patients that would benefit from AR-targeted therapies. Herein, through a review of preclinical studies, clinical studies, and clinical trials, we summarize the biology of AR, its prognostic and predictive value, as well as its therapeutic implications by breast cancer molecular subtype.

## 1. Introduction

Breast cancer (BC) is the most common tumor in the female population [1]. It is characterized by heterogeneity at the molecular and clinical levels. Due to BC complexity, it is no longer considered as a single disease, and the need of classification has arisen. The main BC subtypes have been classified according to their molecular profile and several established biomarkers, including estrogen receptor aplha (ERα), progesterone receptor (PgR), and human epidermal growth receptor-2 (ERBB-2/HER 2) [2,3]. In this vein, BC can be classified into five molecular subtypes, including normal-like, luminal A, luminal B, basal-like, and HER 2-enriched. Molecular classification is vital for the prognosis and prediction of the clinical outcome, as well as the design of the treatment strategy according to patients’ status [4]. For instance, hormone receptor (HR) -positive patients are being treated with hormonal therapy, while HER 2-positive patients are being treated with humanized monoclonal antibodies, exhibiting promising results. On the other hand, cytotoxic chemotherapy is the main systemic therapy for Triple-Negative Breast Cancers (TNBCs), a subset of basal-like tumors which is defined by the lack of therapeutic targets, such as HR and HER 2 [5]. Although, the therapeutic options for specific subtypes of BC seem to be efficient, there is an urgent need to diminish drug resistance and improve clinical benefit by exploring alternative therapeutic targets for this disease. 

Interestingly, it has been noticed that the androgen receptor (AR) is expressed in 70–90% of the BC incidents [6], and that it plays a crucial role in breast cancer pathology and progression. Although AR is implicated in all stages of BC development [7], its function seems to vary among different BC subtypes (Figure 1). AR has been proposed as a potential therapeutic target for BC, and the availability of AR inhibitors approved for prostate cancer treatment could constitute a therapeutic tool for specific subsets of breast cancer.

## 2. Physiology of Androgen Receptors (AR)

The Androgen Receptor (AR) is a type I nuclear receptor. In the absence of a ligand, it is located in the cytoplasm associated with heat shock proteins (HSP) and other chaperons. Once circulating androgens passively diffuse through the cell membrane, they bind to AR causing conformational changes. Subsequently, AR disassociates from heat shock proteins, gets activated, and forms dimers. These dimers translocate to the nucleus, binding to androgen-responsive elements (AREs) within target genes, causing modulations in DNA transcription. AR dimers can regulate gene transcription positively or negatively, leading to differentiation, proliferation, apoptosis, or angiogenesis [8] (Figure 2). AR can also be activated in a ligand-independent manner through cross-talk with key signaling pathways, including PI3K/Akt, ERK, mammalian target of rapamycin (mTOR) and Wnt/β-catenin, or through interaction with other proteins such as Forkhead box Protein A1 (FOXA1) (Figure 1). Non-ERK-mediated AR signaling also involves forkhead box protein O1 (FOXO1) inactivation and protein kinase A (PKA) activation [9,10].

AR is expressed in a variety of tissues in both sexes, including the breast. Interestingly, AR plays a significant role in female biology [11]. In particular, it has been shown that AR is essential for normal fertility in the female population since *AR* knockout mice showed dysfunctional ovulation and impaired follicular growth [12]. Although estrogens play a predominant role in female breast development, androgens are also indispensable in this process. Testosterone in females is synthesized in the ovaries and adrenal glands. In breast tissue, testosterone is converted into dihydrotestosterone (DHT) or 17β-estradiol (E2), and binds onto AR or ERα, leading to the inhibition or stimulation of cell proliferation, respectively [13,14]. Studies have shown that testosterone is preferentially converted into DHT and, with a lack of estrogens, is metabolized into E2, maintaining the hormonal balance within the mammary gland. The importance of functional AR in breast development has been confirmed in in vivo studies where *AR*-knockout mice showed reduced epithelial cell proliferation and reluctant ductal extension in the breast [15]. It has also been shown that androgens inhibit the growth of breast tissue [16], revealing the fundamental role of AR signaling in normal breast tissue development.

## 3. Prognostic Value of AR by Different Breast Cancer Subtypes

The androgen receptor is expressed in more than 70% of primary breast cancers; usually, its expression is correlated to ERα and PgR. AR prevalence is higher in ERα-positive early breast tumors than ERα-negative ones (74.8% vs. 31.8% of cases respectively) [17]. Patients with ERα- and AR-positive tumors have better outcomes than those with ERα-negative/AR-positive status [18,19]. This finding has been attributed to the fact that in ERα-positive BC, AR competes with ERα for the binding to the estrogen-related elements, leading to impaired ERα transcription and apoptosis. In contrast, in ERα-negative tumors, AR binds to androgen-responsive elements (EREs), leading to cell proliferation and tumor growth [20]. These findings have been confirmed by clinical studies, where, among AR-positive patients, ERα positivity was related to better outcome (time to relapse, overall survival) and more favorable clinicopathological features (lower tumor grade, negative lymph node metastasis) [18,19]. Further support has been provided by a retrospective study of 913 patients where tumors with concordant AR and ERα expression had better prognoses than those of discordant AR and ERα (HR = 1.99, 95% CI 1.28–3.10, *p* = 0.02) [20].

A retrospective analysis of 42 Ductal Carcinoma in situ of the breast (DCIS) patients treated with surgery followed by radiotherapy showed that AR expression was considerably higher in relapsed tumors (*p* = 0.0005), whereas ERα was higher in non-relapsed ones. The AR:ERα ratio was different among the subgroups (*p* = 0.0033), indicating the unfavorable prognostic role of AR and AR/ERα in this subset of patients [21]. Consistently, another study revealed that the AR:ERα value could be used as a highly specific and sensitive prognostic tool for in situ relapse or progression to invasive subtypes [22].

Controversial claims about the value of AR on Luminal breast cancers have been reported. In some studies, AR was identified an independent prognostic biomarker when hormone receptors were expressed, whereas in others, AR was shown to be independent from the expression of other hormone receptors [17,23]. Moreover, for Luminal A tumors, it was shown that AR holds a predictive value for positive outcome [24].

In triple negative breast cancer, AR positivity may represent more than 50% of cases, and its expression levels vary considerably among TNBC molecular subtypes [25]. For this subset of BC patients, studies have emerged showing AR to be a solid potential therapeutic target [26]. A retrospective study of 699 patients linked AR-positivity with significantly better disease-free survival [27]. Further studies have associated AR with better overall survival (*p* = 0.04) but lower rates of pathological complete response to neoadjuvant chemotherapy [28]. In luminal androgen receptor (LAR) TNBC, AR positivity was also associated with higher overall survival [29]. A recent clinical study of 135 TNBC patients managed to stratify three different TNBC risk groups with different therapeutic implications. LAR TNBCs (AR-positive, EGFR-negative) belong to the low-risk group with better prognoses and lower proliferation rate. This subgroup might benefit most from antiandrogen targeted therapy. In contrast, AR-negative EGFR-positive TNBCs constitute the high-risk group, with worse prognoses and the highest proliferation rate. This subgroup is expected to benefit from chemotherapy [30]. Taken together, these results show that AR provides new opportunities for the treatment of this subset of breast cancer patients.

## 4. Predictive Role of AR

It seems that the AR:ERα ratio is able predict the response to endocrine therapy [23]. More specifically, a study of 192 patients treated with tamoxifen showed that a high nuclear AR:ERα ratio (at least 2.0 by immunohistochemical staining) could predict failure from hormone therapy. The same study showed that the administration of antiandrogen enzalutamide decreased both ERα-positive and ERα-negative/AR-positive breast tumor growth, suggesting antiandrogen therapy as a novel effective treatment for patients with de novo resistance to hormone therapies [23]. In advanced breast cancers, AR does not seem to predict the efficacy of first-line antiestrogen endocrine therapy, whereas progesterone receptor and Ki67 were shown to be more potent predictors. Time to Progression (TTP) was not significantly associated with AR status, but a ratio of AR:PgR ≥ 0.96 was associated with shorter TTP. Accordingly, in primary tumors and metastases, AR status did not affect the progress of the disease as best response, whereas Ki67 ≥ 20% and PgR < 10% were significantly associated with it [31]. Clinical data support the opinion that AR may not be an informative biomarker for the selection of endocrine therapy. More specifically, the predictive value of AR was evaluated in postmenopausal, ER-positive BC patients who were administrated with either letrozole or tamoxifen monotherapy. The results showed that the treatment effect of neither letrozole nor tamoxifen was associated with the AR status of the tumors (NCT00004205).

On the other hand, plenty of studies have supported the hypothesis that AR status can predict the efficacy of AR inhibitors [32]. In TNBC, AR positivity along with the presence of estrogen receptor beta (ER-β) can predict the efficacy of enzalutamide [33,34]. In ERα-positive/HER 2-positive breast cancers, where AR is expressed in more than 60% of cases [6], studies have suggested that AR positivity is responsible for smaller tumor sizes, lower Ki67 percentages, and less aggressive phenotypes [35]. In contrast, preclinical studies have shown that in ERα-negative, HER 2-positive breast cancers, AR promotes tumor growth through the AR/FOXA1/β-catenin complex which binds to the HER 3 gene, thereby inducing cancer cell proliferation (Figure 1) [10].

## 5. The Therapeutic Potential of AR: Clinical Overview

As discussed, AR can trigger or suppress the oncogenic features of breast tumors depending on the bioavailability of estrogens [36]. In this vain, both AR agonists and antagonists have been tested as potential therapies (Figure 2). Moreover, the interaction of AR with other molecular pathways has led to the investigation of therapies with AR-targeted agents in combination with signal transduction inhibitors.

So far, although, natural and synthetic steroidal androgens have been used as therapeutic approach for ERα-positive BC, they have induced serious side effects [37]. Selective AR modulators (SARMs) such as enobosarm have been preclinically tested giving favorable results concerning migration and invasion. In vivo studies revealed that SARMs were able to reduce the tumor weight by 90%, as well as the tumor-induced cachexia, in 5 weeks [38]. Moreover, postmenopausal women exhibit minor side effects, and the treatment is currently being investigated in a phase II clinical trial for patients with metastatic or locally-advanced ER+ and AR+ breast cancer (NCT02463032) (Table 1). 

AR antagonists are also being investigated in preclinical and clinical studies. Presently-available AR inhibitors are being widely used to treat prostate cancer and are showing encouraging results in several clinical trials in breast cancer. A phase II clinical trial evaluating bicalutamide, a first-generation AR antagonist, in AR-positive/ERα-negative/PgR-negative advanced breast cancers, showed a clinical benefit rate of 19% at 6 months and a median progression-free survival duration of 12 weeks [25]. This was the first proof for efficacy of AR-targeted treatment in AR-positive TNBC. To date, a phase II clinical study testing bicalutamide in metastatic TNBC is ongoing; its outcomes have reported a modest clinical benefit rate of 20% (NCT00468715) (Table 1). 

Enzalutamide is a second-generation AR antagonist approved for castration-resistant prostate cancer patients [39]. A phase II clinical trial evaluating enzalutamide in AR-positive TNBC showed a clinical benefit rate of 33% at 16 weeks, 28% at 24 weeks, median overall survival of 16.5 months, and a median progression free survival duration of 3.3 months in the evaluable subgroup of patients [32]. Another phase II clinical study evaluating enzalutamide in combination with transtuzumab in patients with HER2+ AR+ metastatic or locally-advanced breast cancer (NCT02091960) is ongoing (Table 1). The primary objective was clinical benefit rate at 24 weeks (CBR24), defined as complete or partial response (CR or PR) or stable disease (SD) for ≥24 weeks in evaluable patients. Another ongoing phase IIb trial is investigating whether a combination of enzalutamide and paclitaxel would exhibit benefits in early-stage, AR-positive TNBC patients (NCT02689427) (Table 1). Abiraterone acetate and seviteronel, both CYP17A1 inhibitors, target androgen biosynthesis and production respectively [40], and are currently being clinically evaluated. A phase I/II clinical trial of abiraterone acetate in postmenopausal women with advanced or metastatic breast cancer has been completed, presenting a clinical benefit rate of 20% in 24 weeks with a median progression-free survival duration of 2.8 months (NCT00755885) (Table 1). Another study combining abiraterone acetate with prednisone presented promising results for patients with molecular apocrine-like tumors (NCT01842321) (Table 1) [41]. Seviteronel, which lowers the production of both androgens and estrogens [40], has been evaluated in a phase I/II trial (NCT02580448) (Table 1). Phase I revealed that seviteronel was generally well tolerated in women with TNBC and ER+ breast cancer. Interestingly, the 26.3% and 11% of the subjects reached at least a clinical benefit rate at four (CBR16) and six (CBR24) months, respectively.

At present, the standard treatment for advanced TNBC is exclusively nonspecific, cytotoxic chemotherapy [5]. Although treatments that target AR allow patients with advanced or metastatic TNBC to be treated with less toxic endocrine agents, the need remains for the development of novel predictive biomarkers that will identify the patients that will benefit the most.

## 6. Future Prospects in Prognosis and Therapy

The development of AR testing on liquid biopsy is an active area of research. In prostate cancer, AR concentration in the serum/plasma or urine has already been correlated with diagnosis, prognosis, and outcome prediction [42,43]. Recently, circulating tumor cells (CTCs) in breast cancer were evaluated for the expression of AR-v7 isoform; the results showed a direct association between AR-v7 expression and increased bone metastasis [44]. Further study showed that 31% of CTCs were found to express AR mRNA, the levels of which vary depending on the breast cancer subtype [42]. These findings led to the conclusion that AR levels in CTCs could hold a prognostic value for the progression of the disease, as well as a predictive value for the selection of patients for AR-targeted therapies. Further research on this topic is required.

Studying other prostate cancer research achievements, we can open new horizons in breast cancer research. For instance, the concentration of programmed death-ligand (PD-L1)-positive and PD-L2-positive dendritic cells in circulation may play a role in the selection of BC patients who are resistant to enzalutamide, as has already been shown for PC patients [45].

Moreover, PC research showed that the co-administration of enzalutamide with a therapeutic vaccine targeting the Twist protein, a transcription factor regulating metastasis, achieved overall survival [46]; however, this finding needs to be investigated also in a breast cancer model.

The co-administration of enzalutamide with a therapeutic vaccine targeting the Twist protein in an in vivo prostate cancer model achieved overall survival [46], opening the way for corresponding approaches in breast cancer.

## 7. Conclusions

The value of AR as a biomarker and therapeutic target in breast cancer remains elusive. The contradictory results are due to the heterogeneity of the disease, as well as to the fact that there is no well-defined cut-off value of AR positivity. The dual role of AR as either a suppressor or inducer of tumor progression enables both androgens and antiandrogens to be used in potential therapeutic regiments. Since clinical trials have shown promising results, research should focus on the identification of novel biomarkers in order to stratify patient subgroups that will benefit the most from AR-targeted therapies.

## Figures and Tables

**Figure 1 molecules-25-00358-f001:**
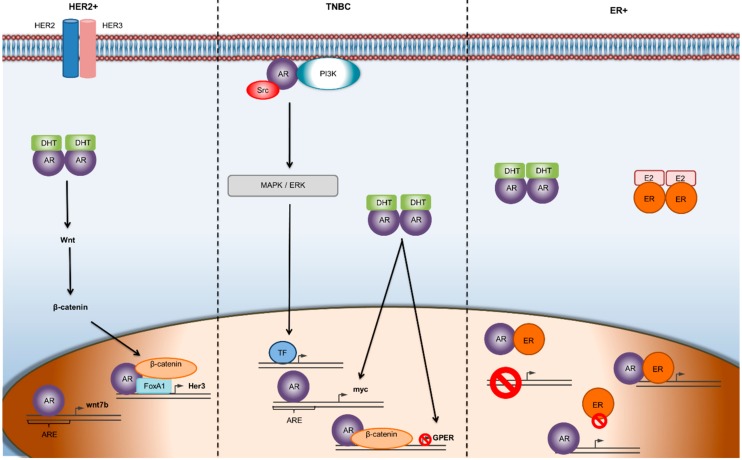
Mechanisms of AR mediated gene transcription in different subtypes of breast cancer. In ER-α-negative/HER 2-positive breast cancer, the Wnt/β-catenin pathway is implicated and facilitates the transcriptional activity of AR promoting tumor growth. In TNBC, androgens seem to initiate second-messenger signaling cascades, which often results in a feedback loop, leading to the progression of the tumor. In the ER-positive, BC subtype, there is a dynamic relationship between ER and AR, where the two receptors can transcriptionally regulate each other through heterodimerization and binding to the same DNA sequence. Abbreviations: AR—androgen receptor; ARE—androgen receptor element; DHT—dihydrotestosterone; TF—transcription factor.

**Figure 2 molecules-25-00358-f002:**
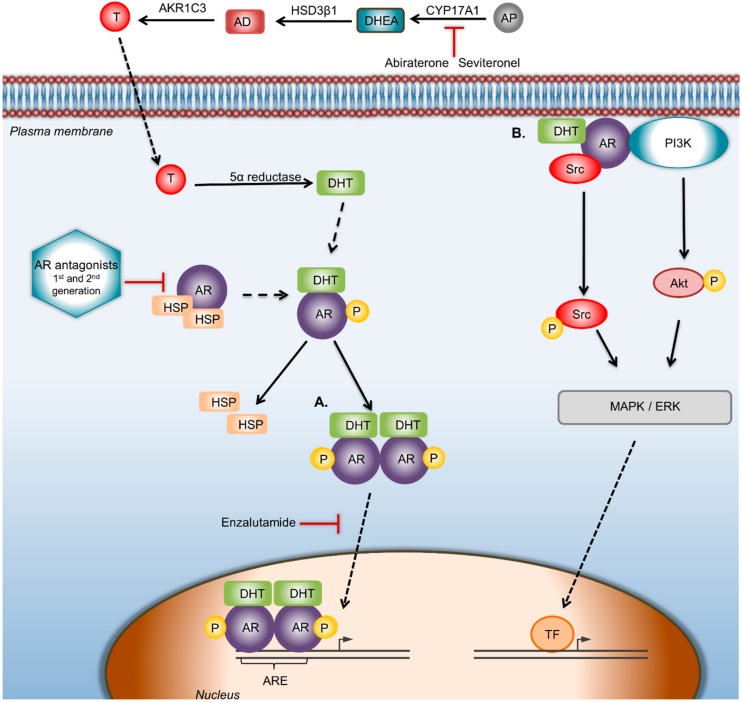
AR signaling pathway and targeted therapeutic strategies against AR. AR can regulate the proliferation, migration, and invasiveness in BC through genomic and/or nongenomic pathways. CYP17A1 is the enzyme that converts androgen precursors into DHEA, HSD3β1 catalyzes DHEA to AD, while AKR1C3 converts AD to testosterone; finally, testosterone is catalyzed to DHT by 5α-reductase. In the genomic way of AR activation, DHT binds and activates AR, which disassociates from heat shock proteins and forms dimers, which translocate to the nucleus, where gene transcription is modulated by binding to the androgen response elements of target genes. In the nongenomic way of AR activation, ERK-mediated AR signaling involves phosphoinositide 3-kinase (PI3K), Src proteins, and Ras GTPase. Abiraterone acetate selectively and irreversibly blocks CYP17A1 activity. Drugs: Seviteronel is a CYP17A1 inhibitor. Bicalutamide is a first-generation and Enzalutamide a second-generation AR antagonist that blocks androgens binding to AR. Enzalutamide also inhibits AR nuclear translocation and AR-mediated transcription. In the nongenomic way of AR activation, the use of AR antagonists/inhibitors of AR-activated proteins that disrupts the AR/src association could be a starting point to reduce BC cell proliferation. Abbreviations: AD—androstenedione; AKR1C3—aldo-keto reductase family 1 member C3; AP—androgen precursors; AR—androgen receptor; ARE—androgen receptor element; CYP17A1—cytochrome P450 c17; DHEA—dehydroepiandrosterone; DHT—dihydrotestosterone; HSP—heat shock protein; HSD3β1—human 3-beta-hydroxysteroid dehydroxynase/delta5-4 isomerase type 1; T—testosterone; TF—transcription factor.

**Table 1 molecules-25-00358-t001:** Androgen blockade-based clinical studies in breast cancer.

Identifier	Study Design	Class of Agents	Agents	Molecular Profile	Patients (*n*)	Endpoint	Status of Trial
NCT02463032	Randomized, Open label Phase II	Selective-AR modulator	GTx-024	ER+ AR+ BC	88	CBR	Ongoing
NCT00468715	Open label Phase II	AR inhibitor	Bicalutamide	Metastatic TNBC	28	CR or PR	Ongoing
NCT02091960	Open label Phase II	AR inhibitor HER 2 Inhibitor	Enzalutamide Transtuzumab	HER2+ AR+ metastatic or locally advanced BC.	103	CBR	Ongoing
NCT00755885	Non randomized Open label Phase I/II	AR inhibitor	Abiraterone Acetate	Advanced or Metastatic AR+ BC	77	MTD CBR	Completed
NCT01842321	Single Group Assignment Open label Phase II	AR inhibitor	Abiraterone Acetate	Advanced or Metastatic TNBC	31	CBR	Ongoing
NCT02580448	Non randomized Open label Phase I/II	Lyase-selective CYP17 inhibitor	Seviteronel	Advanced TNBC, ER+ BC	175	CBR	Completed
NCT02689427	Non randomized Open label Phase IIB	AR Inhibitor Microtubule stabilizer	Enzalutamide Paclitaxel	AR+ TNBC Stage I-III	37	pCR RCB-I	Recruiting
NCT00004205	Randomized, double-blind-phase-III	Aromatase inhibitor Selective ER modulator	Letrozole Tamoxifen	ER+ PgR+ BC	8028	DFS	Completed

Abbreviations: MBC—metastatic breast cancer; CBR—clinical benefit rate; CR or PR—response rate; MTD—maximum tolerated dose; pCR—pathologic complete response; PFS—progress free survival; RCB-I—minimal residual disease; AR+—AR-positive.

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
