# Peer review of "Androgen Receptor in Breast Cancer—Clinical and Preclinical Research Insights"

_molecules, 2020, doi:10.3390/molecules25020358_

Round 1

Reviewer 1 Report

The manuscript “The Androgen Receptor in Breast Cancer” proposes to review an important aspect in breast cancer. The starting point of the study was nice due to the impact of AR in tumorigenesis. However, the manuscript does not present a new approach to the topic. A review article needs to describe a specific issue with a new perspective, and the submitted version does not meet this criterion.

Published articles:

Androgen Receptor Function and Androgen Receptor–Targeted Therapies in Breast Cancer: A Review- JAMA Oncology 3:1266-1273, 2017 (cited by the authors).

The role of the androgen receptor in triple-negative breast cancer - Women's Health 9: 351–360, 2013

Special comments:

1. The authors did not describe properly BC subtypes. Considering AR signaling, the authors, again, should better describe in details AR pathway.

2. A figure should be included to better understand AR mechanism in the breast.

3. The authors cited other review articles, which compromise the quality of the manuscript.

4. “Future Prospects in prognosis and therapy” section must be rewrite. The content has no context and is completely fragmented.

- Positive aspects: the authors cited five new trials with AR agonists and antagonists: NCT02463032; NCT02091960; NCT00755885; NCT01842321; NCT02580448. However, may better describe inclusion and exclusion criteria. The trials may be presented in Table and need a better description.

My recommendation – Considering the need to better conceptualize and contextualize the aspects related to the proposal of the article, especially for clinical trials, the study is premature for publication in Molecules.

Author Response

The manuscript “The Androgen Receptor in Breast Cancer” proposes to review an important aspect in breast cancer. The starting point of the study was nice due to the impact of AR in tumorigenesis. However, the manuscript does not present a new approach to the topic. A review article needs to describe a specific issue with a new perspective, and the submitted version does not meet this criterion.

Published articles:

Androgen Receptor Function and Androgen Receptor–Targeted Therapies in Breast Cancer: A Review- JAMA Oncology 3:1266-1273, 2017 (cited by the authors).

The role of the androgen receptor in triple-negative breast cancer - Women's Health 9: 351–360, 2013

Special comments:

The authors did not describe properly BC subtypes. Considering AR signaling, the authors, again, should better describe in details AR pathway.

We thank the reviewer for their comment. We focused on the three main subtypes of BC (HER2 positive, TNBC and ER positive). Further information about AR signaling have been added along with the two new figures. Figure 1 shows AR signaling in the three different subtypes of BC and figure 2 extensively explains the different ways of AR activation along with the possible therapies.

A figure should be included to better understand AR mechanism in the breast.

We have included a new figure as explained in the comment No1.

The authors cited other review articles, which compromise the quality of the manuscript.

We totally understand the reviewer’s concern. However we feel that the citation of review papers that are considered as milestones in the field, does not compromise the quality of our manuscript.

“Future Prospects in prognosis and therapy” section must be rewrite. The content has no context and is completely fragmented.

We thank the reviewer for their comment. Indeed, several changes have been done on the structure and the context of this section 

- Positive aspects: the authors cited five new trials with AR agonists and antagonists: NCT02463032; NCT02091960; NCT00755885; NCT01842321; NCT02580448. However, may better describe inclusion and exclusion criteria. The trials may be presented in Table and need a better description.

This point has been accommodated according to the reviewer’s comment. We have included table 1. Listing all the needed information for the most important clinical trials on this topic. Moreover, two (2) more clinical trials have been added, in our effort to add further value and clinical direction to our manuscript.

My recommendation – Considering the need to better conceptualize and contextualize the aspects related to the proposal of the article, especially for clinical trials, the study is premature for publication in Molecules.

We hope that the reviewer will now find the revised manuscript suitable for publication.

Reviewer 2 Report

The manuscript "The Androgen Receptor in Breast Cancer" reviews diverse clinical and preclinical data on involvement of AR in different subtypes of the breast cancer. The work is rather a mini-review, which provides a brief overview of the major directions and controversies in the field. Overall, the manuscript is planned rather well, however there are significant concerns which must be addressed:

1) Since this is a rather small review, the title is overall too general and should reflect the content better, e.g. contain some indication that the review is focused on clinical and preclinical results on AR.

2) The figure should illustrate several modalities of AR transcription, i.e. in presence/absence of beta-catenin and it would be nice to put examples of the most crucial genes transcribed, e.g. HER3.

3) The manuscript text contains a large number of mistakes, especially in prepositions (e.g. l28 "in molecular") as well as many others and should be thoroughly proof-read.

Minor concerns:

1) in the section 3, authors cite 2 studies, one which reports 31.8% AR-positive tumors in ER-negative tumors, and the other, which reports >50% AR overexpression in TNBC. These statements are to some degree contradictory and authors should provide a better discussion for this part;

2) l117 - authors discuss therapeutic role of AR in the section called "Predictive role of AR";

3) reference 37 is a general review and should not be used as reference for sentence in line 131;

4) ERBB-2 is a rarely used name for HER2 and it would be better if authors use HER2 at least alongside of ERBB-2

Author Response

The manuscript "The Androgen Receptor in Breast Cancer" reviews diverse clinical and preclinical data on involvement of AR in different subtypes of the breast cancer. The work is rather a mini-review, which provides a brief overview of the major directions and controversies in the field. Overall, the manuscript is planned rather well, however there are significant concerns which must be addressed:

Since this is a rather small review, the title is overall too general and should reflect the content better, e.g. contain some indication that the review is focused on clinical and preclinical results on AR.

We totally agree with the reviewer’s comment and we have changed the title of the review.

The figure should illustrate several modalities of AR transcription, i.e. in presence/absence of beta-catenin and it would be nice to put examples of the most crucial genes transcribed, e.g. HER3.

This point has been accommodated according to the reviewer’s comment by adding figure 1 and figure 2. Figure 1 shows AR signaling in the three different subtypes of BC as well as the transcriptional effect of AR activation and figure 2 extensively explains the different ways of AR activation along with the possible therapies.

The manuscript text contains a large number of mistakes, especially in prepositions (e.g. l28 "in molecular") as well as many others and should be thoroughly proof-read.

We thank the reviewer for this comment. We have made extensive editing to improve the language and the flow of reading.

Minor concerns:

in the section 3, authors cite 2 studies, one which reports 31.8% AR-positive tumors in ER-negative tumors, and the other, which reports >50% AR overexpression in TNBC. These statements are to some degree contradictory and authors should provide a better discussion for this part;

We thank the reviewer for their comment. However, paper cited 25 explains this wide range which “may be attributed to the retrospective nature of these studies and the biases inherent to this type of analysis, variability in patient selection from archival specimens (i.e., primary tumors vs. metastases; coexpression of HER2), differing assays for AR testing, or other factors not yet realized”

2) l117 - authors discuss therapeutic role of AR in the section called "Predictive role of AR";

We thank the reviewer. However we do not actually discuss the therapeutic role of AR. In the previous sentence we mentioned the study which is strongly linked with the predictive role of AR. Following this, we mentioned the additional outcomes of this study.

reference 37 is a general review and should not be used as reference for sentence in line 131;

We agree with the reviewer and we have replaced the citation.

ERBB-2 is a rarely used name for HER2 and it would be better if authors use HER2 at least alongside of ERBB-2

We agree with the reviewer and we have made the corresponding changes.

Round 2

Reviewer 1 Report

The manuscript submitted by Anestis et al. was revised properly and all points were corrected as suggested.
The study might be accepted for publication

Reviewer 2 Report

The authors have correctly and fully answered the raised concerns and strongly improved the manuscript. This work can be now certainly recommended for publication.